# Oviduct Epithelial Cell-Derived Extracellular Vesicles Improve Porcine Trophoblast Outgrowth

**DOI:** 10.3390/vetsci9110609

**Published:** 2022-11-04

**Authors:** Xun Fang, Bereket Molla Tanga, Seonggyu Bang, Chaerim Seo, Heyyoung Kim, Islam M. Saadeldin, Sanghoon Lee, Jongki Cho

**Affiliations:** 1Laboratory of Theriogenology, College of Veterinary Medicine, Chungnam National University, Daejeon 34134, Korea; 2School of Biological Sciences and Technology, College of Natural Sciences, Chonnam National University, Gwangju 34134, Korea; 3Research Institute of Veterinary Medicine, Chungnam National University, Daejeon 34134, Korea

**Keywords:** extracellular vesicles, oviduct epithelial cells, embryonic stem cells, porcine, embryo

## Abstract

**Simple Summary:**

We investigated the effects of extracellular vesicles derived from the oviduct on porcine blastocyst attachment as a model for preimplantation embryo implantation. Embryo developmental competence and pluripotency gene expression were highly expressed after extracellular vesicles supplementation.

**Abstract:**

Porcine species have a great impact on studies on biomaterial production, organ transplantation and the development of biomedical models. The low efficiency of in vitro-produced embryos to derive embryonic stem cells has made achieving this goal a challenge. The fallopian tube plays an important role in the development of embryos. Extracellular vesicles (EVs) secreted by oviductal epithelial cells play an important role in the epigenetic regulation of embryo development. We used artificially isolated oviductal epithelial cells and EVs. In this study, oviductal epithelial cell (OEC) EVs were isolated and characterized through transmission electron microscopy, nanoparticles tracking analysis, western blotting and proteomics. We found that embryo development and blastocyst formation rate was significantly increased (14.3% ± 0.6% vs. 6.0% ± 0.6%) after OEC EVs treatment. According to our data, the inner cell mass (ICM)/trophectoderm (TE) ratio of the embryonic cell number increased significantly after OEC EVs treatment (43.7% ± 2.3% vs. 28.4% ± 2.1%). Meanwhile, the attachment ability of embryos treated with OEV EVs was significantly improved (43.5% ± 2.1% vs. 29.2% ± 2.5%, respectively). Using quantitative polymerase chain reaction (qPCR), we found that the expression of reprogramming genes (POU5F1, SOX2, NANOG, KLF4 and c-Myc) and implantation-related genes (VIM, KRT8, TEAD4 and CDX2) significantly increased in OEC EV-treated embryos. We report that OEC EV treatment can improve the development and implantation abilities of embryos.

## 1. Introduction

Pigs have great clinical value and research significance as animal models and cloning pigs has also become an important research direction in the transplant industry [1,2]. Many endangered species can also reproduce by cloning. However, the low embryo rate and implantation rate of cloned animals have been important obstacles to research and development [3]. Pig embryonic stem cell research can also provide important data for human clinical experience [4,5,6]. The embryo implantation process and its factors are important links to exploration and action [7,8]. Therefore, improving parthenogenetic embryo attachment and pluripotency is a challenge in reproduction.

Embryos require several genes for common regulation and mechanisms during their development. The expression of OCT4, SOX2, NANOG, KLF4 and c-MYC drives the embryo to develop into various early tissues [9]. Simultaneously, reprogrammed gene expression induces epigenetic regulation within the cytoplasm. DNA methylation is a hallmark of cell versatility [10]. Mass opening of the nucleus histones can also greatly increase the activity of the nucleus and increase the level of omnipotent gene expression. Higher quality oocytes have stronger cellular and mitochondrial activity and the embryo’s ability to implant itself varies [11]. Higher expression of reprogrammed genes is necessary to obtain better embryos.

The oviduct tube serves as the “place” where sperm and oocytes are fused. It plays an important role in supporting fertilization and provides a nutrient base for early embryonic development [12,13]. The inner wall of the oviduct tube contains layers of epithelial cells that, when stimulated by hormones, secrete large amounts of fluid, including glucose, pyruvate, glycogen, cytokines, epidermal growth factor (EGF), transforming growth factor (TGF), insulin-like growth factors (IGF), fibroblast growth factor (FGF), embryotropic factor-3, transferrin, albumin, proteins and extracellular vesicles (EVs). Numerous studies have reported that tubal fluid contains approximately 200 proteins. The oviduct-specific glycoprotein (OVGP1) protein, which contains special secretions from fallopian epithelial cells, plays a very important role in the development of embryos. 

Many studies have found that cell-to-cell transmission is not just the transmission of information between cell membranes but can also be between cells in tissue fluid through EVs. EVs are phospholipid bilayer structures in which vacuole structures can carry large amounts of mRNA, miRNA and proteins. There is a great deal of influence on and regulation of epigenetic action between cells. Numerous studies have shown that EVs secreted by oviduct epithelial cells (OECs) have many regulatory mechanisms in the early development of fertilized embryos. In multiple species, including mice [14], cats [15], dogs [16], pigs [17] and cows [18], OEVs can be incorporated into oocytes, sperms and embryos [12].

In this study, exosomes secreted by fallopian epithelial cells were tested for their embryonic development and implantation ability. Our findings will help us better understand the necessity and importance of fallopian tube epithelial cells in embryonic development.

## 2. Materials and Methods

### 2.1. Chemicals

All chemicals and reagents were purchased from Sigma-Aldrich (St. Louis, MO, USA), unless otherwise specified.

### 2.2. Oocyte Collection and In Vitro Maturation (IVM)

In vitro oocyte maturation was performed according to a previously described method [19]. Porcine ovaries were extracted from the local slaughterhouse and delivered to the laboratory in 280 mOsm/L saline at 32–34 °C. Porcine cumulus-oocyte complexes (COCs) were aspirated from large follicles a 10 mL syringe. The follicle fluid (OF) was aspirated from the ovary, transferred into a 50 mL conical tube (SPL, Life Sciences, Seoul, Korea) and spun down for 7 min to isolate the sediments. Excellent quality COCs were collected, washed and selected thrice in porcine HEPES-buffered Tyrode’s medium (TLH) containing 0.05% (*w*/*v*) polyvinyl alcohol (TLH-PVA). The COCs were matured in vitro in medium 199 (TCM-199, Gibco, Fisher Scientific, Waltham, MA, USA) supplemented with 10% (*v*/*v*) porcine follicle fluid (PFF), 0.57 mM L-cysteine, 10 ng/mL EGF, 0.91 mM sodium pyruvate, 1 μg/mL insulin, 10 IU/mL equine chorionic gonadotropin (eCG), 10 IU/mL human chorionic gonadotropin (hCG) and 75 μg/mL kanamycin. We transferred 50 COCs to a 4-well dish (SPL Life Sciences, Seoul, Korea) containing 500 μL of maturation medium. The two-step culture lasted a total of 44 h; during the first 22 h, COCs were cultured with hormones at 38.5 °C in 5% CO_2_. By 22 h, COCs were cultured in a hormone-free IVM medium for an additional 22 h.

### 2.3. Collection of Porcine Oviductal Fluid and Epithelial Cells

Porcine oviduct samples were collected from a local slaughterhouse. Oviducts and ovaries were transported to the laboratory. The samples were kept at 37 °C in saline containing 1% antibiotics for 2–3 h after collection. In all experiments, oviduct tissue was selected at the post-ovulatory stage of the porcine estrous cycle and used for EV collection. To minimize variability, porcine OECs were collected from the same oviduct and used for EV collection. The OECs were collected by the mechanical scraping of the oviduct. Single cells were isolated using collagenase for 1 h. OECs were washed three times in Dulbecco’s phosphate-buffered saline (Gibco) with 1% antibiotic-antimycotic (anti-anti) (Gibco, Billings, MT, USA) and centrifuged at 800× *g* for 3 min. OECs were seeded in a 100 mm tissue culture dish (BD, Franklin Lakes, NJ, USA) with DMEM supplemented with 10% fetal bovine serum (Gibco, Billings, MT, USA) and 1% penicillin/streptomycin mixture (100 U/mL penicillin, 100 µg/mL streptomycin) in a humidified atmosphere of 5% CO_2_ at 39 °C. Realtime PCR and western blot were used to identify the cell line using oviduct-specific glycoprotein (OVGP1) which is expressed by the epithelial cells of the oviduct. When the cell rendezvous rate reached 80%, Fetal Bovine Serum (FBS) was removed from the medium and conditioned media (CM) after 48 h. FBS-free CM was used for EV isolation.

### 2.4. Isolation, Purification and Characterization of EVs

OEC CM was used to isolate EVs using the Total EV Isolation kit (Invitrogen, Carlsbad, CA, USA, Cat # 4478359). Before isolation, the cells and cell debris were removed from the CM and centrifuged at 3000× *g* for 30 min. Cell-free CM was transferred to a 1.5 mL tube with the kit reagent and the samples were vortexed and incubated at 2–8 °C overnight. Overnight samples were centrifuged at 10,000× *g* for 1 h at 2–8 °C. The OEC EV pellets were stored in porcine zygote medium (PZM-5). The protein concentration was adjusted to 50 ng/mL using a NanoDrop 2000 (Thermo Scientific™, Waltham, MA, USA). OEC EV pellets were stored at −80 °C and thawed before the experiments. The resuspended pellets were examined by transmission electron microscopy (TEM) [20]. Then, 10 μL of the EV pellets suspension were loaded on 300-mesh grids. The samples dried 5 min in room temperature. The dried samples were stained with 2% uranyl acetate. EVs were visualized through an energy-filtering TEM (Basic Science Laboratory, Daejeon, Korea) set at 120 kV. The mean diameter of EVs was estimated by ImageJ 1.47t software (National Institutes of Health, Bethesda, MD, USA).

### 2.5. ZetaView Nanoparticle Tracking Analysis (NTA)

A ZetaView PMX 110 (Particle Metrix, Meerbusch, Germany) instrument was used for NTA as previously described [21]. Briefly, 1 mL of diluted sample (in 1× phosphate-buffered saline [PBS]) was loaded into the device and measure each sample at 11 different positions. The all samples were reading 2 cycles per position. After automated analysis and outlier removal, median, the nanoparticle means and mode sizes (indicated as diameter). The EVs concentrations were calculated using ZetaView 8.05.14 SP7 software and Microsoft Excel 365 (Microsoft Corp., Seattle, DC, USA). Device calibration before all samples testing was performed by 100 nm polystyrene particles (ThermoFisher Scientific, Waltham, MA, USA).

### 2.6. Proteomic Analysis of OEC-EVs

Protein profiling of the OEC-EVs was performed according to [22] OEC-EVs pellets contents of protein was estimated analysis through the bicinchoninic acid method and then proteins were fractionated by sodium dodecyl sulfate-polyacrylamide gel electrophoresis (SDS-PAGE). For Coomassie Brilliant Blue staining, the gels were stained by 10 mM ammonium bicarbonate solution and 50% acetonitrile [23]. After the gels were rinsed twice with distilled water and followed by 100% acetonitrile, respectively the gels were dried with a speed vacuum concentrator. The gels were treated with mixture of 100 mM ammonium bicarbonate and 10 mM dithiothreitol at 56 °C, before treatment with 100 nM iodoacetamide to minimize alkylate S–S bridges. The gels were vortexed in three volumes of distilled water for washing and then dried with a speed vacuum concentrator. The gels were incubated in 10 ng/mL trypsin and 50 mM ammonium bicarbonate at 37 °C for 12–16 h for tryptic digestion. Tryptic peptides were retrieved after treatment with 50% acetonitrile containing 5% trifluoroacetic acid and 50 mM ammonium bicarbonate. Peptide extract was lyophilized and stored at 4 °C until further analysis. Tryptic peptide extract was suspended in 0.5% trifluoroacetic acid and 10 μL from each sample was loaded onto MGU30-C18 trapping columns (LC Packings) to concentrate peptides and clear extra chemicals. Concentrated tryptic peptides were eluted from the column and loaded onto a 10 cm × 75 μm I.D. C18 reverse-phase column (PROXEON, Odense, Denmark) at adjusted flow rate (300 nL/min). Peptides were retrieved by a gradient of 0–65% acetonitrile for 80 min. MS and MS/MS spectrum was obtained by using LTQ-Velos ESI ion trap mass spectrometer (Thermo Scientific, Waltham, MA, USA). MASCOT 2.4 was used to analyze MS/MS data with a false discovery rate of 1% as a cutoff value. Protein quantities were estimated through the exponentially modified protein abundance index (emPAI) and were expressed as mol %. Three technical replicates were performed. Functional analysis and gene ontology were performed through the Functional Annotation Tool, DAVID Bioinformatics Resources (NIAID/NIH; https://david.ncifcrf.gov/home.jsp (accessed on 15 September 2022) [24,25].

### 2.7. Western Blot

We performed western blotting according to our previous method [26] with some modifications. Briefly, proteins were extracted from OEC and OEC-derived EVs using Protein Extraction Solution (PRO-PREP^TM^, iNtRON biotechnology, Gyeonggi-do, Korea. 25 µg of POEC and 40 µg of POEC-derived exosomes were added to the sample buffer and mixed well. Diluted samples were loaded into a 15% (*w*/*v*) polyacrylamide gel and electrophoresis was performed for 3 h at a constant voltage (80 V) at room temperature. Precision Plus ProteinTM Standards Dual Color (Bio-Rad, Hercules, CA, USA) was used as a molecular weight marker. The separated proteins were transferred (12 V for O/N at 4 degree) to methanol-activated polyvinylidene difluoride (PVDF) membranes (Immobilon-P transfer membranes, Merck KGaA, Darmstadt, Germany) and blocked for 30 min at room temperature with 5% (*w*/*v*) skimmed milk. Membranes were exposed to mouse monoclonal IgG1 κ β-Actin antibody, mouse monoclonal IgG1 κ CD9 antibody, mouse monoclonal IgG2b κ CD81 antibody (1:1000, Santa Cruz Biotechnology Co., Ltd., Dallas, TX, USA), recombinant Anti-CD63 antibody (1:1000 Abcam, Cambridge, MA, USA) for overnight at 4 °C. After washing 3 times with TBS-T solution. Membranes were exposed to Goat Anti-Rabbit IgG H&L (HRP, ab6721) or Rabbit Anti-Goat IgG H&L (HRP, ab6741) for 2 h. Bands and signals were read using Chemiluminescence imaging-Fusion SOLO software (Vilber Lourmat, France). Band densities were quantified using Image J software (Version 1.41; National Institutes of Health, Bethesda, MD, USA).

### 2.8. Uptake of OEC-EVs by the Preimplantation Embryos

Serum-free conditioned culture medium was blend with the PKH26 lipophilic fluorescent stain following to the manufacturer’s instructions. OEC-EVs were isolated after removing the superfluous PKH67 dye according the manufacturer’s recommendations [27,28].Then, OEC-EVs were supplemented (50 ng/mL) [29] with the embryos for 24 h. The uptake of PKH26-labelled EVs were confirmed through a confocal microscope (Leica DMi8, Wetzlar, Germany). For negative control staining, the blank conditioned medium mixture with PKH26 dye and processed by the same labeling procedure.

### 2.9. Parthenogenetic Activation of Oocytes and Culture

Porcine matured COCs were pipetted with 0.1% hyaluronidase and denuded oocytes were kept in TCM 199 with 0.1% bovine serum albumin (BSA). For parthenogenetic activation, we set two direct-current (DC) pulses of 120 V for 60 μs in an electrically stimulated buffer containing 0.28 M mannitol, 0.1 mM CaCl_2_·2H_2_O and 0.05 mM MgSO_4_·6H_2_O. Electrostimulation was performed using an Electro Cell Manipulator 2001 (BTX, San Diego, CA, USA). The activated oocytes were transferred to a post-activation media drop, PZM-5 containing 7.5 μg/mL cytochalasin B and 4 mg/mL BSA [30] and cultured for 3 h in an incubator. After post-activation, the oocytes were washed thrice with PZM-5. Every ten PA oocytes were cultured in vitro in 25 μL of PZM-5 drops and covered with mineral oil. The embryos were cultured in a humidified atmosphere of 5% CO_2_, 5% O_2_ and 90% N_2_ at 38.5 °C for 7 days.

### 2.10. Preparation of ICR Mouse Feeder Cells

Mouse embryonic fibroblasts (MEFs) used as feeder cell layers were prepared from 13.5 days of pregnancy. All tissues except fibroblast tissue were removed and the remaining fetal tissues were washed three times with PBS (Gibco, Carlsbad, CA, USA) with 1% Anti-anti (Gibco, Carlsbad, CA, USA) and centrifuged at 800× *g* for 2 min. Digestive tissue was digested into homogeneous single cells using trypsin (Gibco, Carlsbad, CA, USA) for 10 min. The MEF were cultured in Dulbecco’s modified Eagle medium (Gibco, Carlsbad, CA, USA) containing 10% FBS, 1% non-essential amino acids (NEAA, Gibco, Carlsbad, CA, USA), 1% Glutamax (Gibco, Carlsbad, CA, USA), 0.1 mM β-mercapto-ethanol and 1% anti-anti (all from Gibco). The MEFs were cultured at 37 °C in a humidified 5% CO_2_ incubator. To produce mitotically inactive ICR MEF feeder cells, mitomycin C (10 µg/mL, Roche, Basel, Switzerland) was added to the MEF for 2–2.5 h. These MEF feeder cells were then plated at a density of 5 × 10^5^ cells/mL in a 6-well dish coated with 0.5% gelatin (WelGENE, Inc., Gyeongsangbuk-do, Korea) containing 2 mL MEF culture medium. Passage 2–3 cells were used for feeding the blastocysts.

### 2.11. Quantitative Real-Time PCR (q-PCR)

Total RNA was extracted from day-7 blastocysts using the RNeasy Micro Kit (QIAGEN, Hilden, Germany) and the RNA was balanced using nanodrops 2000. The isolated RNA was synthesized using 2X RT Pre-Mix (BIOFACT, Daejeon, Korea) to prepare complementary DNA (cDNA) following the product protocol. All primers were designed using BLAST and synthesized as previously reported and all information is presented in (Table 1). The cDNA was run by qPCR with primers in SYBR 2X Real-Time PCR Pre-Mix (BIOFACT, Korea) and the CFX96 real-time PCR detection system (Bio-Rad). The cycling conditions were as per the manufacturer’s protocol. The results of the qPCR data as ΔΔcq were calculated in Excel. Transcripts of the examined genes were quantified in triplicate and calculated as expressions relative to the levels of the housekeeping gene GAPDH in the same sample. Approximately 20 embryos per group were processed for each replicate. Experiments were repeated at least three times.

### 2.12. Statistical Analysis

A minimum of eight replicates were used for each experiment. For data normalization, the average values of expression levels from the control group were calculated and normalized to 1 by dividing the values again. The other experimental groups were compared to the normalized values of the control group. SPSS version 22 (SPSS 22.0; IBM, USA) was used for statistical analysis. Data are expressed as the mean ± standard error of the mean (SEM) and analyzed using the unpaired t-test and univariate analysis variance (ANOVA) with Tukey’s multiple comparison test to determine the significant differences among the experimental groups. *p*-values < 0.05 were considered to be significantly different among the experimental groups.

## 3. Results

### 3.1. EV Isolation, Characterization and Proteomics

Transmission electron microscopy (TEM) images (Figure 1A). results showed the presence of membrane-enveloped vesicles isolated from the conditioned medium of OECs. OEC EVs were characterized by nanoparticles tracking analysis (NTA). Results showed that the finite track length adjustment (FTLA) mean particle size of the isolated OEC EVs was 122.2 ± 1.0 nm and the particle concentrations were 2.46 ± 3.14 particles/mL (Figure 1B). Western blot analysis showed the expression of exosomes markers (CD9, CD63 and CD81) with a highly significant differences with the cell of origin (control) (Figure 1C). Proteomics data of OEC-EVs showed 1041 peptide reads of them 1034 DAVID Uniprot IDs were identified when contrasted to Sus scrofa species (Appendix A). Of these proteins, markers of EVs or exosomes have been identified such as CD9 (A0A5G2R5C9), CD63 (A0A5G2QAW2), CD81 (A0A5G2RLN1) and flotillin (A0A480TGV2). The proteins of OEC-EVs were found to be involved in different functions of the cells such as cell differentiation, transport, adhesion and fertilization (as shown in Figure 2 and Appendix A).

### 3.2. EV Uptake by the Cultured Embryos

The embryonic uptake of Endo-EVs was confirmed by the presence of intracytoplasmic fluorescence signals in 7 days blastocyst. Endo-EVs were labeling with PKH26 stain and their incubation with the embryos for 30 h (Figure 3).

### 3.3. Effect of OEC EVs on Parthenogenetic Embryo Development

We experimentally tested the effect of OEC EVs on embryonic development by using 50 ng/mL OEC EVs to supplement the PZM-5 medium for in vitro culture (IVC) and cultured the parthenogenetically activated oocytes (Table 2). The cleavage rate did not differ between the control and OEC-EV groups. However, the blastocyst and hatching rates were significantly higher in the OEC EV group than in the control group (31.9% ± 0.9% vs. 21.3% ± 0.8% and 14.3% ± 0.6% vs. 6.0% ± 0.6%, respectively).

### 3.4. Effect of OEC EVs on Blastocyst ICM/TE Cell Number

To understand the effect of the OEC EVs on blastocyst hatching, we used OEC EVs to supplement the PZM-5 medium for IVC for 7 days and detected ICM and TE using Hoechst 33342 and propidium iodide. Under the UV microscope, in stained blastocyst the bule color cells was count for ICM and red color cells count for TE cell numbers. The data we obtained according to the differential staining method showed that OEC EVs significantly increase the ICM, TE and ICM/TE rate when compared to the control (15.3 ± 0.6 vs. 8.2 ± 0.5, 33.6 ± 1.2 vs. 29.7 ± 1.5 and 43.7% ± 2.3% vs. 28.4% ± 2.1%, respectively) (Table 3).

### 3.5. Effect of OEC EVs on Blastocyst Attachment

Day 7 PA blastocyst seeding on the ICR MEF was used to check the attachment and extension rates (Figure 4). The OEC EVs group significantly increased the attachment and trophoblast outgrowth rate (43.5% ± 2.1% vs. 29.2% ± 2.5% and 14.1% ± 2.9% vs. 6.2% ± 1.8%, respectively) (Table 4).

### 3.6. Gene Expression of the OEC EVs Treatment Blastocyst

Analysis of gene expression in day-7 blastocysts of the OEC EV and control groups showed the expression were increase of Bcl2, POU5F1, NANOG, SOX2, VIM, c-MYC, Klf4, KRT8, TEAD4 and CDX2, while BAX gene showed significant decrease in the control group (Figure 5).

## 4. Discussion

The pig is the best cloned animal species, but the low implantation ability of cloned embryos has been a major obstacle to its development [31]. There are many differences in the extracellular environment between embryos grown in vitro and those grown in vivo [32]. On one hand, in the early stage of embryo development, cloned embryos grown in vitro have stable MAPK level loss compared to IVF embryos [33,34]. In contrast, the ovum grown in vivo can obtain more abundant cytokines and growth factors and can carry out rich biological information transmission with the epithelial cells of the fallopian tube [35]. Therefore, we believe that in vitro access to bioinformatics that mimic in vivo might lead to significant changes in the ability to produce cloned embryos.

Our study found that EVs isolated from tubal epithelial cells had a significant effect on embryonic development and significant enhancement. Many studies have suggested that EVs can cross the cell membrane and act in the cytoplasm [36,37]. We believe that this is related to epigenetic regulation of embryos by the abundant miRNAs in OEC EVs. Studies have shown that OEC EVs contain a large amount of OVGP1 protein [12]. As a protein secreted by tubal epithelial cells, OVGP1 plays an important role in the early fertilization of oocytes [38]. Relevant data have shown that OEC EVs can greatly improve the success rate of embryo IVF and reduce the rate of multiple sperm entry [39]. Other relevant data indicate that the loss of OEC EVs has a significant impact on embryo development [32]. OEC EVs can reduce the damage caused by reactive oxygen species. It can also increase the level of DNA methylation in embryos [40]. The environment that promotes embryonic development approaches this level in vivo. Our study found that OEC-EVs significantly increased the rate of embryo breakage. This may be related to the improvement in mitochondrial activity and fatty acid metabolism by OEC EVs [40]. Simultaneously, we found that OEC EVs improved the ICM/TE ratio of embryos. Our data showed that OEC EVs could promote the expression of VIM and KRT8 in embryos. High expression of VIM and KRT8 provides cytoskeletal integrity in cells derived from embryonic division [41,42]. We also found that OEC EVs could enhance the expression of reprogramming genes in the ovum. We attempted to evaluate the implantation ability of the OEC EV group. We found that OEV EVs could significantly improve the implantation ability of PA embryos. This may be related to the increased expression of the reprogramming genes. Perhaps miRNAs in OEC EVs as co-promoters can activate the silencing of the reprogramming pathway in the oocyte cytoplasm. Our data suggest that OEC EVs promote the expression of reprogramming genes, such as POU5F1, SOX2, NANOG, KLF4 and C-MYC, in embryos. Reprogramming genes have been implicated in the early implantation of embryos and in the proliferation and differentiation of inner cell masses [43,44]. At the same time, we found that OEC EVs could promote the expression of TEAD4 and CDX2. TEAD4 and CDX2 are important regulatory genes in early embryos that can promote the differentiation of trophoblast cell lineages [45,46,47,48].

Molecular analysis of the EVs cargo contents through proteomics revealed the involvement of the OEC EVs proteins in various physiological processes important in embryonic stem cells adhesion, attachment, migration, cell cycle, metabolism and cell survival (Appendix A).

At present, the mutual regulatory mechanism between OEC EVs and embryos requires further understanding and research. Understanding the mechanism of information transmission and epigenetic regulation will contribute greatly to research on embryo development and the trophoblast outgrowth in the porcine species and can be the starting point for embryonic stem cell research and animal cloning.

## Figures and Tables

**Figure 1 vetsci-09-00609-f001:**
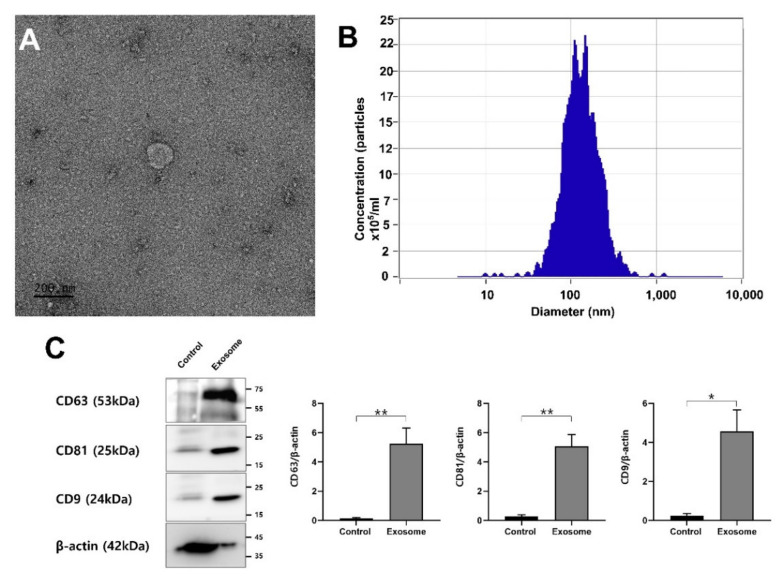
Porcine oviduct epithelial cell derived extracellular vesicles (EVs). (**A**) Transmission electron microscopic (TEM) image of OEC EVs, scale bar = 200 nm. (**B**) Results of nanoparticle tracking analysis (NTA) showing the diameter and concentrations the EVs. (**C**) Western blot results showing the expression of CD9, CD81 and CD63 in both OECs as a control and the OEC-EVs. Values carrying asterisk (*) or (**) are significantly different at *p* < 0.05 or *p* < 0.01, respectively.

**Figure 2 vetsci-09-00609-f002:**
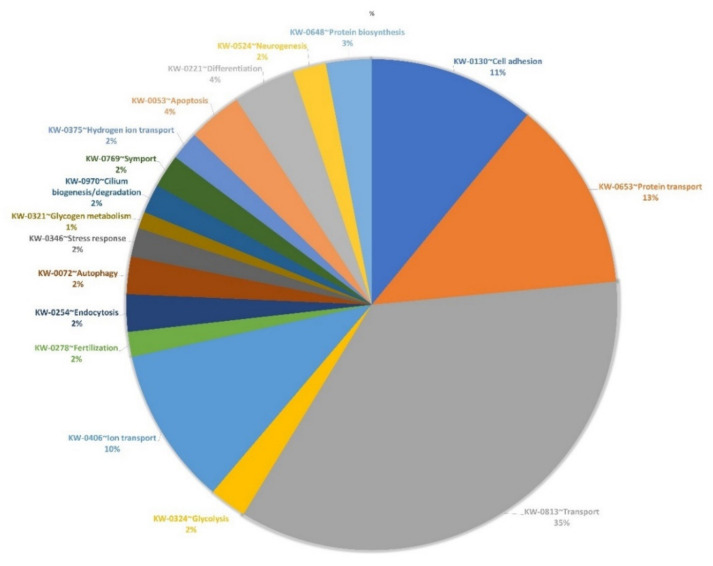
Gene ontology (predicted cellular functions) of the proteins of porcine oviduct epithelial cells derived extracellular vesicles (OEC EVs).

**Figure 3 vetsci-09-00609-f003:**
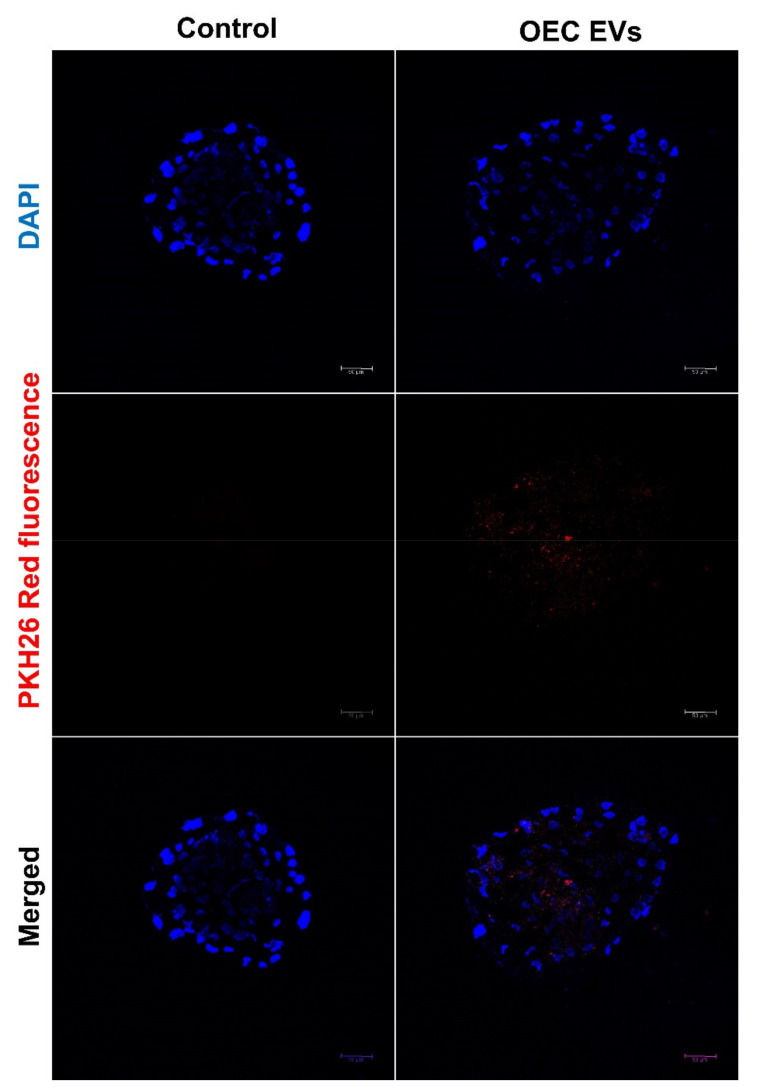
Uptake of OEC EVs by hatched embryos after labelling with lipophilic dye PKH26. Red fluorescent dots were observed after incubation of ethe embryos with the stained EVs. In negative control group, the plain culture medium was handled the same as the EVs group except the presence of EVs.

**Figure 4 vetsci-09-00609-f004:**
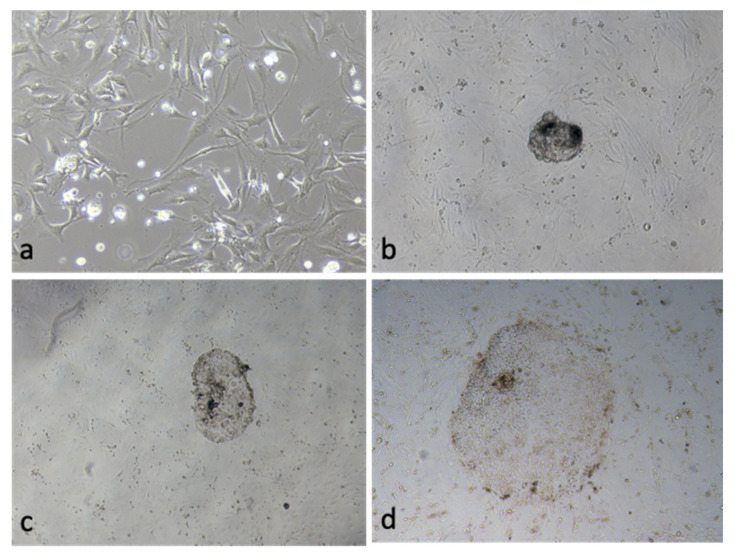
(**a**) ICR mouse embryonic fibroblasts (MEF) feeder cells. (**b**) Seeding of a day-7 porcine parthenogenetically activated blastocyst. (**c**) The blastocyst attachment on feeder cells. (**d**) Trophoblast outgrowth on the ICR MEF feeder cells.

**Figure 5 vetsci-09-00609-f005:**
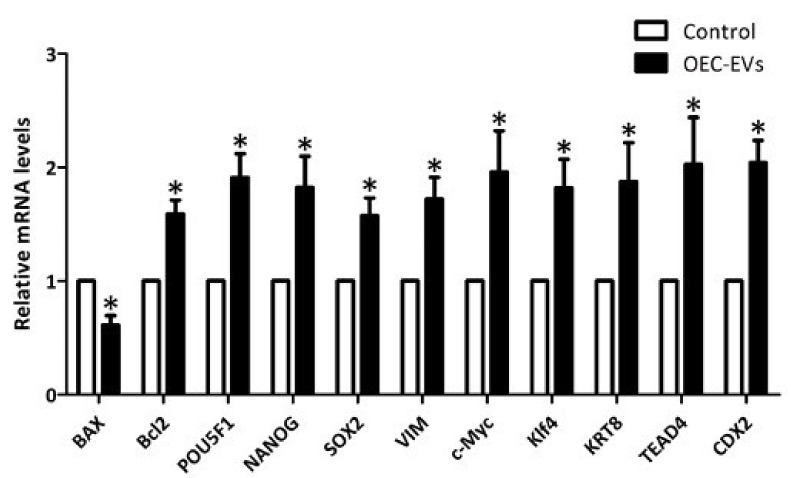
Real-time qPCR analysis of gene expression in the control and oviductal epithelial cell (OEC)-extracellular vesicle (EV) groups in parthenogenetically activated blastocysts (n = 10, 3 replicates). Values carrying asterisk (*) are statistically different at *p* < 0.05.

**Table 1 vetsci-09-00609-t001:** Information on the primer sequences used for real-time PCR analysis.

Genes	Primer Sequences (5′→3′)	Product Size (bp)	Accession No.
GAPDH	F: AGAAGGTGGTGAAGCAGG R: AGCTTGACGAAGTGGTCG	154	NM_001206359.1
BAX	F: ACTTCCTTCGAGATCGGC R: GGCCACGAAGATGGTCAC	110	XM_003127290
Bcl-2	F: TTCTCTCGTCGCTACCGC R: CCAGTTCACCCCATCCCT	123	XM_021099593
POU5F1	F: GCCAGAAGGGCAAACGAT R: AGGGTGGTGAAGTGAGGG	154	NM_001113060
NANOG	F: AGACTTGGAATAGCCAGC R: CCGCAGTACTTTGAAGTC	151	NM_001129971
SOX2	F: TACAGCATGATGCAGGAC R: GAGCTGGTCATGGAGTTG	128	NM_001123197
VIM	F: CAGGCTCAGATCCAGGAACA R: GTCGGCAAACTTGGACTTGT	156	XM_005668107.3
c-Myc	F: CAACGTCAGCTTCACCAACA R: TGGGCAGCAACTCGAATTTC	164	NM_001005154.1
Klf4	F: CCAAACTACCCACCCTTCCT R: CTAGGGGTGAAGAAGGTGGG	163	XM_005660316.3
KRT8	F: AAGCTGGTGTCTGAGTCCTC R: GAATTGGCTTGGAGTTGGGG	166	NM_001159615.1
TEAD4	F: ATCGGATGAGGGCAAGATGT R: GCCTGATCCTTTAGCTTGGC	159	NM_001142666.1
CDX2	F: GTGTTAAACCCCACCGTCACR: CAACCGCACCTGTCTTTACC	194	NM_001278769.1

GAPDH: Glyceraldehyde-3-phosphate dehydrogenase; BAX: BCL2 associated X; Bcl-2: B-cell lymphoma 2; POU5F1: POU class 5 homeobox 1; NANOG: Nanog homeobox; SOX2: SRY-box transcription factor 2; VIM: vimentin; c-Myc: MYC proto-oncogene; Klf4: Kruppel-like factor 4; KRT8: Keratin 8; TEAD4: TEA domain transcription factor 4; CDX2: caudal type homeobox 2.

**Table 2 vetsci-09-00609-t002:** Effect of OEC EVs during in vitro culture of porcine oocyte and developmental potential after parthenogenesis.

Groups	No. of Embryos			
	Cultured	Cleaved (% ± SEM)	Develop to BL (% ± SEM)	Hatched BL (% ± SEM)
Control	554	472 (85.2 ± 0.6)	118 (21.3 ± 0.8)	33 (6.0 ± 0.6)
OEC EVs	573	495 (86.4 ± 0.9)	183 (31.9 ± 0.9) *	82 (14.3 ± 0.6) *

Values carrying asterisk (*) in the same column are significantly different (*p* < 0.05).

**Table 3 vetsci-09-00609-t003:** Comparison of the Control and OEC-EVs treatment blastocyst ICM/TE cell number.

Groups	No. of Cells		
	ICM ± SEM	TE ± SEM	ICM/TE % ± SEM
Control	8.2 ± 0.5	29.7 ± 1.5	28.4 ± 2.1
OEC-EVs	15.3 ± 0.6 *	33.6 ± 1.2 *	43.7 ± 2.3 *

Values carrying asterisk (*) in the same column are significantly different (*p* < 0.05).

**Table 4 vetsci-09-00609-t004:** Effect of OEC EVs on blastocyst attachment on the MEF cells.

Groups	No. of Cells		
	Cultured BL	Attachment (% ± SEM)	Outgrowth (% ± SEM)
Control	65	19 (29.2 ± 2.5)	4 (6.2 ± 1.8)
OEC-EVs	85	37 (43.5 ± 2.1) *	12 (14.1 ± 2.9) *

Values carrying asterisk (*) in the same column are significantly different (*p* < 0.05).

## Data Availability

The data that support the findings of this study will be available from the corresponding author upon a reasonable request.

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
