# Peer review of "Oviduct Epithelial Cell-Derived Extracellular Vesicles Improve Porcine Trophoblast Outgrowth"

_vetsci, 2022, doi:10.3390/vetsci9110609_

Round 1

Reviewer 1 Report

The manuscript by Fang et al. entitled “Oviduct epithelial cell-derived extracellular vesicles improve porcine trophoblast outgrowth” describes the effect of extracellular vesicles secreted by porcine oviductal epithelial cells. Generally the manuscript is interesting, however some aspect should be clarified.

1. line 70: “We aimed to investigate how to increase the embryo development rate and implant ability in animals”. This is not the direct aim of the study. It should be rewritten.

2. There is lack of statement about the approval of ethic committee regarding use of animals.

3. line 107: What “1% anti-anti” means?

4. The catalog number of Total EV Isolation kit should be added.

5. Figure2b: What does a pick around 1000 nm means?

6. The uptake of EVs by oviductal cells should be confirmed.

Author Response

The manuscript by Fang et al. entitled “Oviduct epithelial cell-derived extracellular vesicles improve porcine trophoblast outgrowth” describes the effect of extracellular vesicles secreted by porcine oviductal epithelial cells. Generally the manuscript is interesting, however some aspect should be clarified.

R. We acknowledge the efforts, comments, and suggestions of the reviewer and we considered all these comments and suggestions that greatly contributed to improve the quality of the manuscript.

  1. line 70: “We aimed to investigate how to increase the embryo development rate and implant ability in animals”. This is not the direct aim of the study. It should be rewritten.

R1. We deleted this sentence.

  1. There is lack of statement about the approval of ethic committee regarding use of animals.

R2. We thank the reviewer for this comment. Our study was exempted from the ethical approval committee in our college because the entire experiments were in vitro and we obtained the ovaries from a commercial slaughterhouse and no one from the authors has been involved in the animal handling or slaughtering.

  1. line 107: What “1% anti-anti” means?

R3. We corrected this accordingly.

  1. The catalog number of Total EV Isolation kit should be added.

R4. The Catalog number was added.

  1. Figure2b: What does a pick around 1000 nm means?

R5. We deleted this part as it may confuse the reader.

  1. The uptake of EVs by oviductal cells should be confirmed.

R6. We added the uptake of PKH26 labelled OEC-EVs in the text.

Reviewer 2 Report

Authors have suggested that oviduct epithelial cell-derived extracellular vesicles improve porcine trophoblast outgrowth. The study is on an interesting note but also lacks several aspects.

1. It could have been better to show markers that are specific to oviduct epithelial cell-derived extracellular vesicles to clearly distinguish it avoiding that it is not the artifact. In current figure 1, it is a bit difficult to say that these blebs are actually the oviduct epithelial cell-derived extracellular vesicles.

2. To improve the report better, I would suggest including western blot quantification as well. It is just difficult to believe only q-PCR results about the expression of progenitor cell markers.

Author Response

Authors have suggested that oviduct epithelial cell-derived extracellular vesicles improve porcine trophoblast outgrowth. The study is on an interesting note but also lacks several aspects.

R. We acknowledge the efforts, comments, and suggestions of the reviewer and we considered all these comments and suggestions that greatly contributed to improve the quality of the manuscript.

  1. It could have been better to show markers that are specific to oviduct epithelial cell-derived extracellular vesicles to clearly distinguish it avoiding that it is not the artifact. In current figure 1, it is a bit difficult to say that these blebs are actually the oviduct epithelial cell-derived extracellular vesicles.

R1. We fully understand the reviewer’s meaning. The current work is a continuation of our series of experiments (Fang et al. 2022, Molecular Reproduction and Development). In our early experiments, we isolated those epithelial cells by physical and enzymatic treatment. We have used qPCR and western blot to identify the cell line using oviduct-specific glycoprotein (OVGP1) which is expressed by the epithelial cells of the oviduct, and there is no problem with cell morphology. Moreover, we recently performed proteomics analysis for further characterization, and we added the detailed description of the EVs contents and importantly the markers of exosomes or EVs are expressed in the proteomes data such as CD9, CD63, CD81, and flotillin. We also performed western blot for the EVs markers such as CD9, CD63, and CD81 and showed that in the revised manuscript.

  1. To improve the report better, I would suggest including western blot quantification as well. It is just difficult to believe only q-PCR results about the expression of progenitor cell markers.

R2. We thank the reviewer for this suggestion. As this work is a brief report, we will consider this analysis in our future experiments. Currently we have another PhD student who works on the effects of different steroid hormone treatments on the oviduct functions and the yield of their EVs a well as the subsequent effects on embryo development.

Reviewer 3 Report

The article “Oviduct epithelial cell-derived extracellular vesicles improve porcine trophoblast outgrowth“ brought interesting results about effect of porcine oviduct extracellular vesicles on developmental potential of parthenotes and also their effect on transcriptomic profile. The English of the manuscript needs correction. For the reasons given below, I recommend a major revision.

General comment: EVs isolates should be analyzed by western blotting with antibodies against exosome-specific markers to confirm present of EVs (positive signal) and also by antibodies against cells-specific markers to confirm that cells are not present (negative signal).

Line 48: Higher quality oocytes...

Line 51: Not fertilized, but fused.

Line 194: In Abstract, you said that there is difference in cleavage of embryos, but in Results you said that not. So how it is?

Author Response

The article “Oviduct epithelial cell-derived extracellular vesicles improve porcine trophoblast outgrowth“ brought interesting results about effect of porcine oviduct extracellular vesicles on developmental potential of parthenotes and also their effect on transcriptomic profile. The English of the manuscript needs correction. For the reasons given below, I recommend a major revision.

R. We acknowledge the efforts, comments, and suggestions of the reviewer and we considered all these comments and suggestions that greatly contributed to improve the quality of the manuscript. We revised the English writing throughout the manuscript.

1. General comment: EVs isolates should be analyzed by western blotting with antibodies against exosome-specific markers to confirm present of EVs (positive signal) and also by antibodies against cells-specific markers to confirm that cells are not present (negative signal).

R1. We fully understand the reviewer’s meaning. The current work is a continuation of our series of experiments (Fang et al. 2022, Molecular Reproduction and Development). In our early experiments, we isolated those epithelial cells by physical and enzymatic treatment. We have used qPCR and western blot to identify the cell line using oviduct-specific glycoprotein (OVGP1) which is expressed by the epithelial cells of the oviduct, and there is no problem with cell morphology. Moreover, we recently performed proteomics analysis for further characterization, and we added the detailed description of the EVs contents and importantly the markers of exosomes or EVs are expressed in the proteomes data such as CD9, CD63, CD81, and flotillin. We also performed western blot for the EVs markers such as CD9, CD63, and CD81 and showed that in the revised manuscript.

2. Line 48: Higher quality oocytes...

R2. Corrected.

3. Line 51: Not fertilized, but fused.

R3. Corrected.

4. Line 194: In Abstract, you said that there is difference in cleavage of embryos, but in Results you said that not. So how it is?

R4. We are sorry for this error. We edited the text accordingly.

Round 2

Reviewer 1 Report

All suggestions have been implemented.

Reviewer 3 Report

Changes in manuscript are reflecting my previous concerns. I recommend to accept it in present form.